# Enhancing Psychotherapy Counseling: A Data Augmentation Pipeline Leveraging Large Language Models for Counseling Conversations

## Abstract

We introduce a pipeline that leverages Large Language Models (LLMs) to transform single-turn psychotherapy counseling sessions into multi-turn interactions. While AI-supported online counseling services for individuals with mental disorders exist, they are often constrained by the limited availability of multi-turn training datasets and frequently fail to fully utilize therapists' expertise. Our proposed pipeline effectively addresses these limitations. The pipeline comprises two main steps: 1) Information Extraction and 2) Multi-turn Counseling Generation. Each step is meticulously designed to extract and generate comprehensive multi-turn counseling conversations from the available datasets. Experimental results from both zero-shot and few-shot generation scenarios demonstrate that our approach significantly enhances the ability of LLMs to produce higher quality multi-turn dialogues in the context of mental health counseling. Our pipeline and dataset are publicly available here.

## 1 Introduction

In contemporary society, the prevalence of mental illness is rising, requiring expert counseling. As large language models (LLMs) like OpenAI's ChatGPT demonstrate ease of access and potential for counseling, increasing numbers of people are interested in using these tools as private counselors [Lin et al., 2023; Choudhury et al., 2023].

However, existing AI-assisted chatbot services, typically designed for everyday conversation, often fall short due to inadequate training and fail to match the quality of responses provided by human experts. Another challenge is the limited availability of multi-turn counseling datasets. Bots trained on single-turn interactions are often incapable of offering practical solutions. While single-turn datasets are more readily available, there is a scarcity of high-quality multi-turn counseling data. Existing multi-turn psychotherapy counseling datasets, such as HOPE [Malhotra et al., 2021] and MEMO [Srivastava et al., 2022], are valuable resources. However, these datasets have not fully utilized the unique counseling styles that every expert possesses. The effectiveness of psychological counseling is enhanced when both the counselor and the client have a better understanding of each other's information. This can be understood as the creation of high-quality counseling data when such information is provided.

To address aforementioned challenges, we propose a pipeline that generates multi-turn counseling conversations based on the client's and psychotherapist's information. Recent research has shown that LLMs can be applied to data augmentation tasks [Zheng et al., 2023; Dai et al., 2023]. These models exhibit excellent capabilities in generating synthetic text data that reflect various generation conditions. Therefore, we utilize this ability to augment the source dataset. Our objective is to integrate experts' distinct counseling styles into the construction of a synthetic multi-turn counseling dataset, resulting in a resource that is both more realistic and practically applicable.

Our contributions are threefold:

- We present a novel pipeline for augmenting psychotherapy multi-turn counseling data augmentation using LLMs. This pipeline leverages the characteristics of both clients and psychotherapists to generate practical counseling data.

- We release an augmented dataset of multi-turn counseling chat that incorporates details of the client's mental illness and the psychotherapist's counseling characteristics.

- We demonstrate the effectiveness of our data augmentation pipeline in enhancing the performance of LLMs.

## 2 Related Work

The application of LLMs in psychological counseling has garnered significant interest in recent years. Several studies are being conducted to enable LLMs to replace the role of counselors in psychological counseling [Chung et al., 2023; Liu et al., 2023a; Fu et al., 2023; Lai et al., 2023].

Building on the foundational advancements in LLMs, recent research has increasingly focused on the development of specialized datasets to enhance the performance of LLMs in psychological counseling. The effectiveness of these models in counseling applications is significantly influenced by the quality and relevance of the training data. Therefore, creating specialized datasets that encapsulate various aspects of psychological counseling is crucial for improving the models' ability to generate contextually appropriate and em-

pathetic responses [Inaba *et al.*, 2024; Qiu *et al.*, 2024; Chen *et al.*, 2023; Li *et al.*, 2024; Bertagnolli, 2020]. High-quality datasets in psychotherapy counseling can be likened to transcripts of effective psychotherapy conversations. Research suggests that effective psychotherapy stems from a good relationship between the psychotherapist and the client [Herman, 1998; Tschuschke *et al.*, 2022].

Our research extends these efforts by proposing a pipeline for transforming single-turn psychological counseling data[Bertagnolli, 2020] into high-quality multi-turn datasets and validating the effectiveness of the constructed data. We adopt a multi-faceted approach that includes collecting information from both counselors and clients, as well as gathering data on various mental health disorders.

## 3 Preliminary

### 3.1 Task Definition

The augmentation is based on the source dataset from CounselChat[1]. We selected this dataset as our source data because it includes information such as responses from multiple therapists to the same counseling session and the preference voting results for those responses. Given the source dataset $D_i = (x_i, y_i, m_i)$, where $x_i$ represents client's utterance, $y_i$ represents therapist's response to the client's utterance and $m_i$ represents client's mental disorder. Our aim is to augment the source data $D_i$ into a multi-turn dataset $D_i' = (x_i', y_i', m_i, c_i, t_i)$. Here, $x_i' = (x_i^1, x_i^2, \ldots, x_i^k)$ represents the augmented client's utterances derived from the base $x_i$ into $k$ multi-turns. $y_i' = (y_i^1, y_i^2, \ldots, y_i^k)$ represents the augmented therapist's responses derived from the base $y_i$ into $k$ multi-turns. Therefore, the augmented $x_i^1$ and $y_i^1$ are $x_i$ and $y_i$ from the source data, respectively. Additionally, $c_i$ contains the client's information, and $t_i$ is the therapist's counseling characteristics both of which are extracted from the source dataset.

### 3.2 Source Dataset Pre-processing

To maximize the utilization of therapist information, we preprocess the source dataset. According to the latest South Korean government report[2], which identifies severe stress, continuous depression, anxiety, and sleep disorders, as the most prevalent mental illnesses in 2022, we have selected these conditions as our focus for an augmented dataset.

We hypothesize that therapists with more recommendations have provided appropriate and beneficial responses to clients. Therefore, we select the expert who has received the most recommendations among the responses of various experts to the client utterance previously identified for the mental disorder. To verify the simplicity and feasibility of our approach, we decided to focus on a single expert per client utterance. Since the information is extracted from therapists who have provided good responses to specific mental disorder-related questions, this extracted therapist information can be

valuable for data augmentation. The characteristics of the selected therapists' counseling sessions are utilized to augment the source dataset into a high-quality multi-turn dataset.

## 4 Method

We propose an augmentation pipeline for extending single-turn psychotherapy counseling data into multi-turn counseling dialogue. The pipeline is shown in Figure 1. Our pipeline consists of two steps: 1) Information extraction and 2) Multi-turn dialogue generation. In the second phase, all four sub-prompts are used to generate multi-turn counseling dialogue. Among various publicly available LLMs, we utilize the instruction-tuned version of Llama3-70B [AI@Meta, 2024] to generate multi-turn dialogue. Each step will be explained in detail in the following sections.

### 4.1 Information Extraction

For the first step, we focus on extracting key information from the source data. Given the client's utterance $x_i$, the therapist's response $y_i$, and the client's mental disorder $m_i$, we extract the inherent information from both the client and therapist in the single-turn dialogue. Additionally, we extract a description of the client's mental disorder. This information is crucial for deriving multi-turn dialogues from the source data, enabling for the construction of more realistic dialogues.

### 4.2 Multi-turn Counseling Generation

In this step, we construct a prompt to extend single-turn counseling into multi-turn counseling dialogue. It is composed of four sub prompts: 1) Description prompt, 2) Condition prompt, 3) Information Prompt, and 4) Answer Prompt.

**Description Prompt**

The description prompt outlines an overview of the content.

> The following is a transcript of a chat between a psychotherapist and a client about {client's mental disorder}.

In the {client's mental disorder} field, we place $m_i$ from the source dataset.

**Condition Prompt**

The condition prompt sets the guidelines for the conditions that the model should follow.

> The client starts the conversation as [client] and the psychotherapist starts the conversation as [psychotherapist]. Please use the dialog and speakers info as a guide to continue your consultation like #format#. Never create anything other than the #format# and just complete the "utterance" part.

This clarifies who initiates the conversation, what information the model should utilize, and the generation format the model should follow.

---

[1]https://towardsdatascience.com/counsel-chat-bootstrapping-high-quality-therapy-data-971b419f33da

[2]https://www.ncmh.go.kr/mentalhealth/main.do

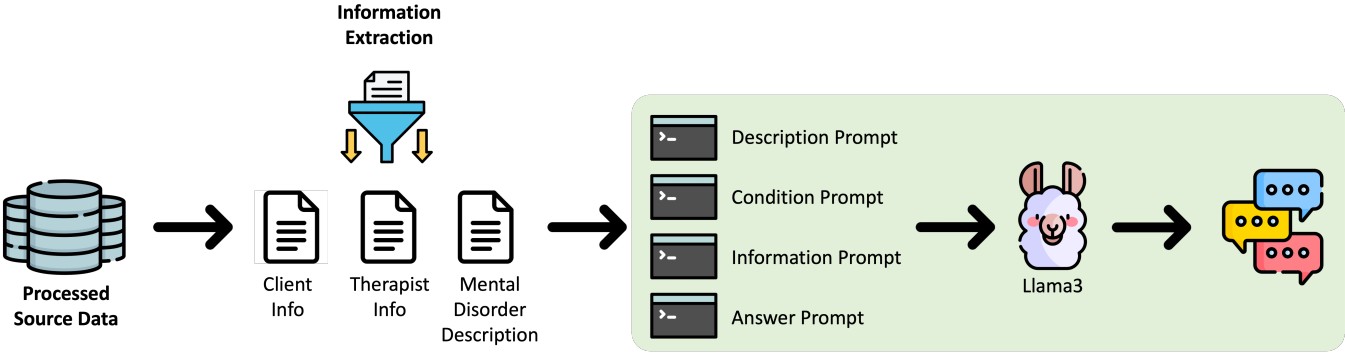

Figure 1: Overview of the proposed data augmentation pipeline.

**Information Prompt**

Using the information extracted from Section 4.1, we first provide the client's mental disorder information. This allows the model to generate a counseling conversation that takes into account the client's symptoms. Then, we include a single-turn, $x_i$ and $y_i$, from the source data to direct the model on what it should generate.

**Answer Prompt**

In the answer prompt, we instruct the model on the proper format for generating the conversation.

```
#format#
[client]:"utterance"
[psychotherapist]: "utterance"
```

The model should generate appropriate utterances in the "utterance" field. This structured format reduces confusion when the model generates its answer and aids in the post-processing of generated text.

## 5 Augmented Dataset

The augmented dataset's distribution of mental disorder categories, generated by our pipeline, is presented in Table 1, which lists the number of cases for each disorder.

Table 1: Mental disorder categories of augmented data

| Mental Disorder of Client | Number of Cases |
|---|---|
| Depression | 69 |
| Anxiety | 45 |
| Anger Management | 16 |
| Trauma | 13 |

Depression is the most prevalent, with 69 cases, indicating it is the most common issue addressed. Anxiety follows with 45 cases, showing a significant presence but less than Depression. Anger Management and Trauma are less common, with 16 and 13 cases, respectively. This distribution reflects the original category distribution of the source dataset.

## 6 Experiment

We conduct zero-shot and few-shot experiments to demonstrate the effectiveness of our pipeline for generating synthetic multi-turn counseling conversations. For the zero-shot experiment, we did not incorporate any specific psychological counseling data into the model and only configured the psychotherapy role prompts. In contrast, for the few-shot experiment, we utilized the dataset we constructed as input. The specific examples used in the actual experiments are detailed in section 6.1 .

### 6.1 Experiment Details

**Test Dataset**

We select 70 dialogues from augmented data randomly. Table 2 presents the mental disorder categories of clients in the test data.

Table 2: Mental disorder categories of test data

| Mental Disorder of Client | Number of Cases |
|---|---|
| Depression | 34 |
| Anxiety | 22 |
| Anger Management | 8 |
| Trauma | 6 |

**Baseline models**

We use Llama2-7B, Llama3-70B as baseline models for generating multi-turn counseling dialogues. We chose these models because they are open-sourced and easily reproducible. We also compare these results with the Llama2-7B model that has been fine-tuned on the Counsel Chat dataset, which is available on HuggingFace [3].

**Experimental setting**

We evaluate two experimental settings: 1) zero-shot multi-turn dialogue generation (see Figure 2 for example), 2) few-shot multi-turn dialogue generation (see Figure 3 for ex-

---

[3]https://huggingface.co/NadunAnjanaka/Llama-2-7b-chat-Counsellor

ample). Results are compared within the same mental disorder category. We consider these settings because well-constructed synthetic multi-turn counseling data should improve the model's few-shot dialogue generation capabilities.

> The following is a transcript of a chat between a psychotherapist and a client about depression. The client starts the conversation as [client] and the psychotherapist starts the conversation as [psychotherapist]. Please complete new transcript about [Question].
>
> [Question]
> [client] I'm almost never happy. Half of the time, I don't feel anything. I find it easy to make myself feel nothing. I know I push people away because it's easier. I just want answers. I'm sick of feeling this way. It's ruining my relationships with people.
> [psychotherapist]

Figure 2: Example of zero-shot prompt

### Automatic Evaluation

We conduct an automatic evaluation using GPT-4o, the latest model introduced by OpenAI. The evaluation prompt is described in Figure 4
We follow the G-Eval methodology [Liu *et al.*, 2023b] for conducting automatic evaluations.

## 6.2  Result

### Evaluation of Zero-shot and Few-shot Performance

Table 3 presents the average scores evaluated by GPT-4o for the two models, Llama2-7B-Chat and Llama3-70B-Instruct, in zero-shot and few-shot settings. The average score in the zero-shot category is 3.814 for Llama2-7B-Chat, slightly lower than the 4.042 for Llama3-70B-Instruct, indicating marginally better performance by Llama3-70B-Instruct when no prior examples are provided. In the few-shot setting, Llama2-7B-Chat has an average score of 4.557, while Llama3-70B-Instruct achieves 4.785, demonstrating that both models perform better when examples generated by our pipeline are provided.

Table 3: Average evaluation score evaluated by GPT-4o

| Model | Zeroshot Avg. | Fewshot Avg. |
|---|---|---|
| Llama2-7B-Chat | 3.814 | 4.557 |
| Llama2-70B-Instruct | 4.042 | 4.785 |

Figure 5 compares the win rates of two models, Llama2-7B-chat and Llama3-70B-Instruct, in terms of their few-shot multi-turn generation performance. Zero-shot wins are shown in green, few-shot wins in peach, and ties in light grey. Each count reflects the win rate comparison between the two models' zero-shot and few-shot capabilities. Llama2-7B-chat recorded 14 zero-shot wins, 55 few-shot wins, and

> The following is a transcript of a chat between a psychotherapist and a client about depression. The client starts the conversation as [client] and the psychotherapist starts the conversation as [psychotherapist]. Please use the following [Example] as a guide complete new transcript about [Question].
>
> [Example]
> [client] They don't go away, and I feel like I'm going crazy. Does that ever stop? Can it be a symptom of medication?
> [psychotherapist] Since you realize that hearing voices in your head is not usual for you, then definitely there is a problematic situation happening within your awareness of who you are.if you recently started taking a new drug or increased dosage of one you already were taking, and the voices started shortly after, then yes, it is possible medication created your problem.Start by telling whoever gave you the presecription, about the problem you're having."Crazy" has some flexibility as to whether someone is this way or not.Certainly a very positive sign that you're not crazy, is that you're self-aware of a problem within yourself. And, you're responsible toward yourself and making effort to address this problem.Crazy people usually don't do responsible behaviors.
> [client] I've been taking the same medication for a while now, but the dosage was increased a few weeks ago. Could that be the cause of the voices?
> [psychotherapist] That's a good point. The dosage increase could definitely be a contributing factor. It's possible that your body is reacting to the higher dosage in a way that's causing these symptoms. I would still recommend reporting this to your prescribing doctor, as they can help you determine the best course of action.
>
> [Question]
> [client] I have been dealing with depression and anxiety for a number of years. I have been on medication, but lately my depression has felt worse. Can counseling help?
> [psychotherapist]

Figure 3: Example of few-shot prompt

1 Tie. Llama3-70B-Instruct, on the other hand, had 11 zero-shot wins, 56 few-shot wins, and 3 ties. The results suggest that Llama3-70B-Instruct performs slightly better in few-shot multi-turn generation. In both Llama2 and Llama3 model, using few-shot examples constructed by our pipeline contributes to generating better quality of multi-turn counseling conversation. Using few-shot examples constructed by our pipeline enhances the models' ability to generate high-quality multi-turn counseling conversations in both Llama2

You're an assistant who evaluates answers strictly from the psychotherapist's perspective about {mental disorder category}. Please rate [Answer 1] and [Answer 2] for the consultation [Question], respectively. Rate the two answers on a scale of 1-5, with higher values indicating better answers.

Figure 4: Example of evaluation prompt

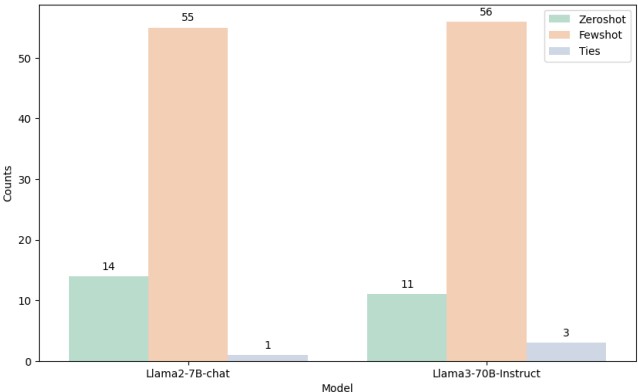

Figure 5: Comparison of zero-shot and few-shot multi-turn counseling dialogue generation performance for Llama2-7B-chat and Llama3-70B-Instruct. In the few-shot setting, examples generated by our pipeline are used.

and Llama3.

## Evaluation of Zero-shot and Few-shot Performance based on Mental Disorder Categories

Figure 6 shows a comparative analysis of win counts by category for Llama2-7B-Chat (upper) and Llama3-70B-Instruct (lower). Win counts are categorized into zero-shot (green), few-shot (peach), and tie (grey) for mental disorders including Depression, Anxiety, Anger-management, and Trauma. Llama2-7B-Chat demonstrates the highest win counts in the few-shot category, with 28 wins in Depression and 15 in Anxiety. Similarly, Llama3-70B-Instruct shows robust few-shot performance, recording 27 wins in Depression and 18 in Anxiety. This comparison shows that both models are particularly effective in few-shot scenarios across all disorders. The overall trend highlights the superior performance of few-shot dialogue generation for both models in addressing various psychological disorders.

## Evaluation of Performance Based on Data Used

Table 4: Win rate and average evaluation score evaluated by GPT-4o

| Model | Win rate | Avg. |
|---|---|---|
| Llama2-7B-Chat-Counsellor[Bertagnolli, 2020] | 0.014 | 1.828 |
| Llama2-7B-Chat Few-shot | 0.985 | 4.528 |

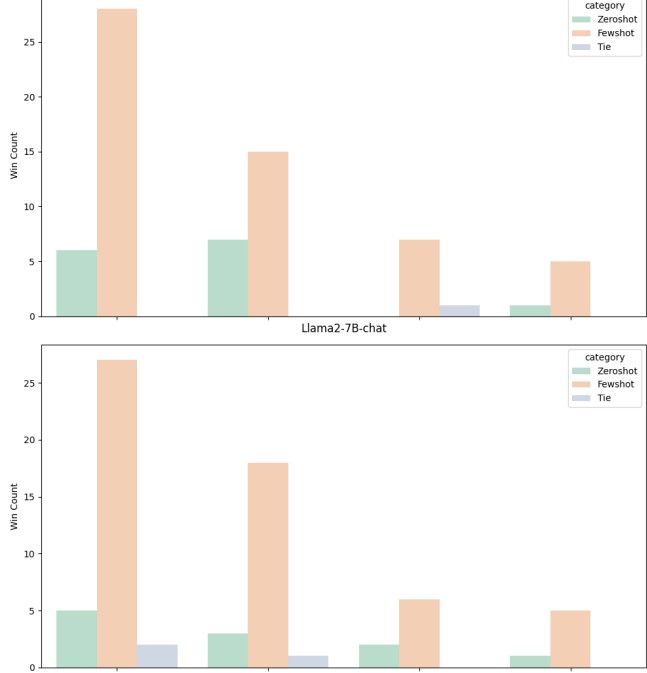

Figure 6: Comparison of zero-shot and few-shot multi-turn counseling dialogue generation performance across mental disorder categories for Llama2-7B-Chat and Llama3-70B-Instruct.

The experimental setup involved comparing Llama2-7B-Chat-Counsellor and Llama2-7B-Chat Few-shot, with the results illustrated in Table 4.

Table 4 demonstrates a clear comparison between the two models' performance. Llama2-7B-Chat-Counsellor achieved an average score of 1.828, while Llama2-7B-Chat Few-shot significantly outperformed Llama2-7B-Chat-Counsellor with an average score of 4.528. These results indicate that Llama2-7B-Chat Few-shot is markedly more effective in generating high-quality, contextually appropriate, and supportive responses in extended multi-turn interactions compared to Llama2-7B-Chat-Counsellor.

This substantial difference in average scores underscores the effectiveness of the pipeline proposed in our research for transforming single-turn data into high-quality multi-turn datasets. The enhanced performance of Llama2-7B-Chat Few-shot suggests that the comprehensive approach we employed, which includes collecting detailed information from both counselors and clients and incorporating data on various mental health disorders, is crucial for improving the capabilities of LLMs in psychological counseling scenarios.

## 7 Conclusion

In this paper, we experimentally assess the utility of the pipeline we propose by comparing multi-turn psychotherapy counseling dialogues generated using our pipeline with those generated by models trained solely on original data. We demonstrate that extracting implicit information from the original data to use as input for multi-turn generation aids in

producing high-quality multi-turn dialogues. This is analogous to the way effective counseling, which alleviates clients' mental disorders, occurs when therapists and clients have a mutual understanding of each other's situations. This experiment provides practical implications not only for researchers and practitioners in the psychotherapy domain but also for those exploring domains with limited multi-turn dialogue data, offering a baseline pipeline for data augmentation.

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
