# OpenReview forum: "Enhancing Psychotherapy Counseling: A Data Augmentation Pipeline Leveraging Large Language Models for Counseling Conversations"
_ijcai.org/IJCAI/2024/Workshop/AI4Research — AI4Research 2024_

### Official Review · Reviewer_8JHy · 2024-06-01
**This paper was interesting which leveraged LLMs to data augmentation for psychotherapy counseling and was well written.**

**Rating:** 7
**Confidence:** 5

**Review:**

This paper proposed a data augmentation pipeline that leverages LLMs to transform single-turn psychotherapy counseling sessions into multi-turn interactions, which could obtain the higher quality by the experimental results from both zero-shot and few-shot generation scenarios. This idea was not novel but interesting for psychotherapy counseling. The experiments were sufficient. Please check the format of the references, especially the case, such as "AugGPT" in Line 338.

---

### Official Review · Reviewer_ctTt · 2024-06-02
**A LLM Data Augmentation pipeline for Counseling Conversations generation**

**Rating:** 5
**Confidence:** 4

**Review:**

This paper proposes a Data Augmentation pipeline to generate multi-turn counseling conversations in Psychotherapy Counseling. It first extract information for the prompting in a therapist data. And then it use Llama3 model with Zero-shot and Few-shot prompts to generate augmented dataset and test dataset. Finally it use  a automatic evaluation by GPT-4o. The result shows using the augmented data with LLMs is better than use the original data with LLMs.

The paper is well-written and clear. It develops a new pipeline for augmenting psychotherapy multi-turn counseling data. The author claims that the augmented data is important for psychotherapists. However, the contributions are limited to Psychotherapy Counseling.

But the paper has some weaknesses. The biggest problem is that it does not include other multi-turn dialogue generation works in NLP, making it too narrow in the Psychotherapy Counseling field. The ethical issue is part of the paper: all the works in this paper, including Information Extraction, Data Generation, and Automatic Evaluation, are based on computer algorithms. However, using these in the medical field like Psychotherapy Counseling could raise reliability and safety issues.

There are also some minor problems:

1. It does not compare the Llama3 model to other instruction-tuned models like GPT models.
2. tt does not have an ablation study for the prompts.
3. It does not compare the pipeline with related works.
4. It does not introduce the criteria for the evaluation of the augmented data and how to split it into Test data.
5. The evaluation score should be explained more clearly.

Reference related works:
A Survey on Recent Advances in LLM-Based Multi-turn Dialogue Systems Yi at al., 2024
Dialog generation using multi-turn reasoning neural networks, Wu et al., 2020
Multi-turn dialogue response generation in an adversarial learning framework, Olabiyi et al., 2018

---

### Decision · Program_Chairs · 2024-06-03

Accept